# On the Power of Truncated SVD for General High-rank Matrix Estimation Problems

**Simon S. Du**
Carnegie Mellon University
ssdu@cs.cmu.edu

**Yining Wang**
Carnegie Mellon University
yiningwa@cs.cmu.edu

**Aarti Singh**
Carnegie Mellon University
aartisingh@cmu.edu

## Abstract

We show that given an estimate $\widehat{\mathbf{A}}$ that is close to a general high-rank positive semi-definite (PSD) matrix $\mathbf{A}$ in spectral norm (i.e., $\|\widehat{\mathbf{A}} - \mathbf{A}\|_2 \leq \delta$), the simple truncated Singular Value Decomposition of $\widehat{\mathbf{A}}$ produces a *multiplicative* approximation of $\mathbf{A}$ in *Frobenius* norm. This observation leads to many interesting results on general high-rank matrix estimation problems:

1. *High-rank matrix completion*: we show that it is possible to recover a general high-rank matrix $\mathbf{A}$ up to $(1 + \varepsilon)$ relative error in Frobenius norm from partial observations, with sample complexity independent of the spectral gap of $\mathbf{A}$.

2. *High-rank matrix denoising*: we design an algorithm that recovers a matrix $\mathbf{A}$ with error in Frobenius norm from its noise-perturbed observations, without assuming $\mathbf{A}$ is exactly low-rank.

3. *Low-dimensional approximation of high-dimensional covariance*: given $N$ i.i.d. samples of dimension $n$ from $\mathcal{N}_n(\mathbf{0}, \mathbf{A})$, we show that it is possible to approximate the covariance matrix $\mathbf{A}$ with relative error in Frobenius norm with $N \approx n$, improving over classical covariance estimation results which requires $N \approx n^2$.

## 1 Introduction

Let $\mathbf{A}$ be an unknown general high-rank $n \times n$ PSD data matrix that one wishes to estimate. In many machine learning applications, though $\mathbf{A}$ is unknown, it is relatively easy to obtain a crude estimate $\widehat{\mathbf{A}}$ that is close to $\mathbf{A}$ in spectral norm (i.e., $\|\widehat{\mathbf{A}} - \mathbf{A}\|_2 \leq \delta$). For example, in matrix completion a simple procedure that fills all unobserved entries with 0 and re-scales observed entries produces an estimate that is consistent in spectral norm (assuming the matrix satisfies a spikeness condition, standard assumption in matrix completion literature). In matrix de-noising, an observation that is corrupted by Gaussian noise is close to the underlying signal, because Gaussian noise is isotropic and has small spectral norm. In covariance estimation, the sample covariance in low-dimensional settings is close to the population covariance in spectral norm under mild conditions [Bunea and Xiao, 2015].

However, in most such applications it is not sufficient to settle for a spectral norm approximation. For example, in recommendation systems (an application of matrix completion) the zero-filled re-scaled rating matrix is close to the ground truth in spectral norm, but it is an absurd estimator because most of the estimated ratings are zero. It is hence mandatory to require a more stringent measure of performance. One commonly used measure is the *Frobenius norm* of the estimation error $\|\widehat{\mathbf{A}} - \mathbf{A}\|_F$, which ensures that (on average) the estimate is close to the ground truth in an element-wise sense. A spectral norm approximation $\widehat{\mathbf{A}}$ is in general *not* a good estimate under Frobenius norm, because in high-rank scenarios $\|\widehat{\mathbf{A}} - \mathbf{A}\|_F$ can be $\sqrt{n}$ times larger than $\|\widehat{\mathbf{A}} - \mathbf{A}\|_2$.

In this paper, we show that in many cases a powerful multiplicative low-rank approximation in Frobenius norm can be obtained by applying a simple truncated SVD procedure on a crude, easy-to-find spectral norm approximate. In particular, given the spectral norm approximation condition $\|\widehat{\mathbf{A}} - \mathbf{A}\|_2 \le \delta$, the top-$k$ SVD of $\widehat{\mathbf{A}}_k$ of $\widehat{\mathbf{A}}$ multiplicatively approximates $\mathbf{A}$ in Frobenius norm; that is, $\|\widehat{\mathbf{A}}_k - \mathbf{A}\|_F \le C(k, \delta, \sigma_{k+1}(\mathbf{A}))\|\mathbf{A} - \mathbf{A}_k\|_F$, where $\mathbf{A}_k$ is the best rank-$k$ approximation of $\mathbf{A}$ in Frobenius and spectral norm. To our knowledge, the best existing result under the assumption $\|\widehat{\mathbf{A}} - \mathbf{A}\|_2 \le \delta$ is due to Achlioptas and McSherry [2007], who showed that $\|\widehat{\mathbf{A}}_k - \mathbf{A}\|_F \le \|\mathbf{A} - \mathbf{A}_k\|_F + \sqrt{k}\delta + 2k^{1/4}\sqrt{\delta\|\mathbf{A}_k\|_F}$, which depends on $\|\mathbf{A}_k\|_F$ and is not multiplicative in $\|\mathbf{A} - \mathbf{A}_k\|_F$.

Below we summarize applications in several matrix estimation problems.

**High-rank matrix completion**    Matrix completion is the problem of (approximately) recovering a data matrix from very few observed entries. It has wide applications in machine learning, especially in online recommendation systems. Most existing work on matrix completion assumes the data matrix is *exactly* low-rank [Candes and Recht, 2012, Sun and Luo, 2016, Jain et al., 2013]. Candes and Plan [2010], Keshavan et al. [2010] studied the problem of recovering a low-rank matrix corrupted by stochastic noise; Chen et al. [2016] considered sparse column corruption. All of the aforementioned work assumes that the ground-truth data matrix is exactly low-rank, which is rarely true in practice.

Negahban and Wainwright [2012] derived minimax rates of estimation error when the spectrum of the data matrix lies in an $\ell_q$ ball. Zhang et al. [2015], Koltchinskii et al. [2011] derived oracle inequalities for general matrix completion; however their error bound has an additional $O(\sqrt{n})$ multiplicative factor. These results also require solving computationally expensive nuclear-norm penalized optimization problems whereas our method only requires solving a single truncated singular value decomposition. Chatterjee et al. [2015] also used the truncated SVD estimator for matrix completion. However, his bound depends on the nuclear norm of the underlying matrix which may be $\sqrt{n}$ times larger than our result. Hardt and Wootters [2014] used a "soft-deflation" technique to remove condition number dependency in the sample complexity; however, their error bound for general high-rank matrix completion is additive and depends on the "consecutive" spectral gap $(\sigma_k(\mathbf{A}) - \sigma_{k+1}(\mathbf{A}))$, which can be small in practical settings [Balcan et al., 2016, Anderson et al., 2015]. Eriksson et al. [2012] considered high-rank matrix completion with additional union-of-subspace structures.

In this paper, we show that if the $n \times n$ data matrix $\mathbf{A}$ satisfies $\mu_0$-spikeness condition, [1] then for any $\epsilon \in (0,1)$, the truncated SVD of zero-filled matrix $\widehat{\mathbf{A}}_k$ satisfies $\|\widehat{\mathbf{A}}_k - \mathbf{A}\|_F \le (1 + O(\epsilon))\|\mathbf{A} - \mathbf{A}_k\|_F$ if the sample complexity is lower bounded by $\Omega(\frac{n \max\{\epsilon^{-4}, k^2\}\mu_0^2\|\mathbf{A}\|_F^2 \log n}{\sigma_{k+1}(\mathbf{A})^2})$ ,which can be further simplified to $\Omega(\mu_0^2 \max\{\epsilon^{-4}, k^2\}\gamma_k(\mathbf{A})^2 \cdot nr_s(\mathbf{A}) \log n)$, where $\gamma_k(\mathbf{A}) = \sigma_1(\mathbf{A})/\sigma_{k+1}(\mathbf{A})$ is the $k$th-order condition number and $r_s(\mathbf{A}) = \|\mathbf{A}\|_F^2/\|\mathbf{A}\|_2^2 \le \mathrm{rank}(\mathbf{A})$ is the *stable rank* of $\mathbf{A}$. Compared to existing work, our error bound is multiplicative, gap-free, and the estimator is computationally efficient. [2]

**High-rank matrix de-noising**    Let $\widehat{\mathbf{A}} = \mathbf{A} + \mathbf{E}$ be a noisy observation of $\mathbf{A}$, where $\mathbf{E}$ is a PSD Gaussian noise matrix with zero mean and $\nu^2/n$ variance on each entry. By simple concentration results we have $\|\widehat{\mathbf{A}} - \mathbf{A}\|_2 = \nu$ with high probability; however, $\widehat{\mathbf{A}}$ is in general not a good estimator of $\mathbf{A}$ in Frobenius norm when $\mathbf{A}$ is high-rank. Specifically, $\|\widehat{\mathbf{A}} - \mathbf{A}\|_F$ can be as large as $\sqrt{n}\nu$.

Applying our main result, we show that if $\nu < \sigma_{k+1}(\mathbf{A})$ for some $k \ll n$, then the top-$k$ SVD $\widehat{\mathbf{A}}_k$ of $\widehat{\mathbf{A}}$ satisfies $\|\widehat{\mathbf{A}}_k - \mathbf{A}\|_F \le (1 + O(\sqrt{\nu/\sigma_{k+1}(\mathbf{A})}))\|\mathbf{A} - \mathbf{A}_k\|_F + \sqrt{k}\nu$. This suggests a form of bias-variance decomposition as larger rank threshold $k$ induces smaller bias $\|\mathbf{A} - \mathbf{A}_k\|_F$ but larger variance $k\nu^2$. Our results generalize existing work on matrix de-noising [Donoho and Gavish, 2014, Donoho et al., 2013, Gavish and Donoho, 2014], which focus primarily on exact low-rank $\mathbf{A}$.

**Low-rank estimation of high-dimensional covariance**   The (Gaussian) covariance estimation problem asks to estimate an $n \times n$ PSD covariance matrix $\mathbf{A}$, either in spectral or Frobenius norm, from $N$ i.i.d. samples $X_1, \cdots, X_N \sim \mathcal{N}(\mathbf{0}, \mathbf{A})$. The *high-dimensional* regime of covariance estimation, in which $N \approx n$ or even $N \ll n$, has attracted enormous interest in the mathematical statistics literature [Cai et al., 2010, Cai and Zhou, 2012, Cai et al., 2013, 2016]. While most existing work focus on sparse or banded covariance matrices, the setting where $\mathbf{A}$ has certain low-rank structure has seen rising interest recently [Bunea and Xiao, 2015, Kneip and Sarda, 2011]. In particular, Bunea and Xiao [2015] shows that if $n = O(N^\beta)$ for some $\beta \geq 0$ then the sample covariance estimator $\widehat{\mathbf{A}} = \frac{1}{N} \sum_{i=1}^N X_i X_i^\top$ satisfies

$$\|\widehat{\mathbf{A}} - \mathbf{A}\|_F = O_{\mathbb{P}}\left( \|\mathbf{A}\|_2 r_e(\mathbf{A}) \sqrt{\frac{\log N}{N}} \right), \tag{1}$$

where $r_e(\mathbf{A}) = \mathrm{tr}(\mathbf{A})/\|\mathbf{A}\|_2 \leq \mathrm{rank}(\mathbf{A})$ is the *effective rank* of $\mathbf{A}$. For high-rank matrices where $r_e(\mathbf{A}) \approx n$, Eq. (1) requires $N = \Omega(n^2 \log n)$ to approximate $\mathbf{A}$ consistently in Frobenius norm.

In this paper we consider a reduced-rank estimator $\widehat{\mathbf{A}}_k$ and show that, if $\frac{r_e(\mathbf{A}) \max\{\epsilon^{-4}, k^2\} \gamma_k(\mathbf{A})^2 \log N}{N} \leq c$ for some small universal constant $c > 0$, then $\|\widehat{\mathbf{A}}_k - \mathbf{A}\|_F$ admits a relative Frobenius-norm error bound $(1 + O(\epsilon))\|\mathbf{A} - \mathbf{A}_k\|_F$ with high probability. Our result allows reasonable approximation of $\mathbf{A}$ in Frobenius norm under the regime of $N = \Omega(n\,\mathrm{poly}(k) \log n)$ if $\gamma_k = O(\mathrm{poly}(k))$, which is significantly more flexible than $N = \Omega(n^2 \log n)$, though the dependency of $\epsilon$ is worse than [Bunea and Xiao, 2015]. The error bound is also agnostic in nature, making no assumption on the actual or effective rank of $\mathbf{A}$.

**Notations**   For an $n \times n$ PSD matrix $\mathbf{A}$, denote $\mathbf{A} = \mathbf{U} \boldsymbol{\Sigma} \mathbf{U}^\top$ as its eigenvalue decomposition, where $\mathbf{U}$ is an orthogonal matrix and $\boldsymbol{\Sigma} = \mathrm{diag}(\sigma_1, \cdots, \sigma_n)$ is a diagonal matrix, with eigenvalues sorted in descending order $\sigma_1 \geq \sigma_2 \geq \cdots \geq \sigma_n \geq 0$. The spectral norm and Frobenius norm of $\mathbf{A}$ are defined as $\|\mathbf{A}\|_2 = \sigma_1$ and $\|\mathbf{A}\|_F = \sqrt{\sigma_1^2 + \cdots + \sigma_n^2}$, respectively. Suppose $\boldsymbol{u}_1, \cdots, \boldsymbol{u}_n$ are eigenvectors associated with $\sigma_1, \cdots, \sigma_n$. Define $\mathbf{A}_k = \sum_{i=1}^k \sigma_i \boldsymbol{u}_i \boldsymbol{u}_i^\top = \mathbf{U}_k \boldsymbol{\Sigma}_k \mathbf{U}_k^\top$, $\mathbf{A}_{n-k} = \sum_{i=k+1}^n \sigma_i \boldsymbol{u}_i \boldsymbol{u}_i^\top = \mathbf{U}_{n-k} \boldsymbol{\Sigma}_{n-k} \mathbf{U}_{n-k}^\top$ and $\mathbf{A}_{m_1:m_2} = \sum_{i=m_1+1}^{m_2} \sigma_i \boldsymbol{u}_i \boldsymbol{u}_i^\top = \mathbf{U}_{m_1:m_2} \boldsymbol{\Sigma}_{m_1:m_2} \mathbf{U}_{m_1:m_2}^\top$. For a tall matrix $\mathbf{U} \in \mathbb{R}^{n \times k}$, we use $\mathcal{U} = \mathrm{Range}(\mathbf{U})$ to denote the linear subspace spanned by the columns of $\mathbf{U}$. For two linear subspaces $\mathcal{U}$ and $\mathcal{V}$, we write $\mathcal{W} = \mathcal{U} \oplus \mathcal{V}$ if $\mathcal{U} \cap \mathcal{V} = \{\mathbf{0}\}$ and $\mathcal{W} = \{\boldsymbol{u} + \boldsymbol{v} : \boldsymbol{u} \in \mathcal{U}, \boldsymbol{v} \in \mathcal{V}\}$. For a sequence of random variables $\{X_n\}_{n=1}^\infty$ and real-valued function $f : \mathbb{N} \to \mathbb{R}$, we say $X_n = O_{\mathbb{P}}(f(n))$ if for any $\epsilon > 0$, there exists $N \in \mathbb{N}$ and $C > 0$ such that $\Pr[|X_n| \geq C \cdot |f(n)|] \leq \epsilon$ for all $n \geq N$.

## 2   Multiplicative Frobenius-norm Approximation and Applications

We first state our main result, which shows that truncated SVD on a weak estimator with small approximation error in spectral norm leads to a strong estimator with *multiplicative* Frobenius-norm error bound. We remark that truncated SVD in general has time complexity

$$O\left( \min\left\{ n^2 k, \mathrm{nnz}\left(\widetilde{\mathbf{A}}\right) + n\,\mathrm{poly}(k) \right\} \right),$$

where $\mathrm{nnz}(\widetilde{\mathbf{A}})$ is the number of non-zero entries in $\widetilde{\mathbf{A}}$, and the time complexity is at most linear in matrix sizes when $k$ is small. We refer readers to [Allen-Zhu and Li, 2016] for details.

**Theorem 2.1.**   *Suppose $\mathbf{A}$ is an $n \times n$ PSD matrix with eigenvalues $\sigma_1(\mathbf{A}) \geq \cdots \geq \sigma_n(\mathbf{A}) \geq 0$, and a symmetric matrix $\widehat{\mathbf{A}}$ satisfies $\|\widehat{\mathbf{A}} - \mathbf{A}\|_2 \leq \delta = \epsilon^2 \sigma_{k+1}(\mathbf{A})$ for some $\epsilon \in (0, 1/4)$. Let $\mathbf{A}_k$ and $\widehat{\mathbf{A}}_k$ be the best rank-$k$ approximations of $\mathbf{A}$ and $\widehat{\mathbf{A}}$. Then*

$$\|\widehat{\mathbf{A}}_k - \mathbf{A}\|_F \leq (1 + 32\epsilon)\|\mathbf{A} - \mathbf{A}_k\|_F + 102\sqrt{2k}\epsilon^2 \|\mathbf{A} - \mathbf{A}_k\|_2. \tag{2}$$

**Remark 2.1.**   *Note when $\epsilon = O(1/\sqrt{k})$ we obtain an $(1 + O(\epsilon))$ error bound.*

**Remark 2.2.**   *This theorem only studies PSD matrices. Using similar arguments in the proof, we believe similar results for general asymmetric matrices can be obtained as well.*

To our knowledge, the best existing bound for $\|\widehat{\mathbf{A}}_k - \mathbf{A}\|_F$ assuming $\|\widehat{\mathbf{A}} - \mathbf{A}\|_2 \leq \delta$ is due to Achlioptas and McSherry [2007], who showed that

$$
\begin{aligned}
\|\widehat{\mathbf{A}}_k - \mathbf{A}\|_F &\leq \|\mathbf{A} - \mathbf{A}_k\|_F + \|(\widehat{\mathbf{A}} - \mathbf{A})_k\|_F + 2\sqrt{\|(\widehat{\mathbf{A}} - \mathbf{A})_k\|_F \|\mathbf{A}_k\|_F} \\
&\leq \|\mathbf{A} - \mathbf{A}_k\|_F + \sqrt{k}\delta\|\mathbf{A} - \mathbf{A}_k\|_2 + 2k^{1/4}\sqrt{\delta}\sqrt{\|\mathbf{A}_k\|_F}.
\end{aligned}
\tag{3}
$$

Compared to Theorem 2.1, Eq. (3) is not relative because the third term $2k^{1/4}\sqrt{\|\mathbf{A}_k\|_F}$ depends on the $k$ *largest* eigenvalues of $\mathbf{A}$, which could be much larger than the remainder term $\|\mathbf{A} - \mathbf{A}_k\|_F$. In contrast, Theorem 2.1, together with Remark 2.1, shows that $\|\widehat{\mathbf{A}}_k - \mathbf{A}\|_F$ could be upper bounded by a small factor multiplied with the remainder term $\|\mathbf{A} - \mathbf{A}_k\|_F$.

We also provide a gap-dependent version.

**Theorem 2.2.** *Suppose $\mathbf{A}$ is an $n \times n$ PSD matrix with eigenvalues $\sigma_1(\mathbf{A}) \geq \cdots \geq \sigma_n(\mathbf{A}) \geq 0$, and a symmetric matrix $\widehat{\mathbf{A}}$ satisfies $\|\widehat{\mathbf{A}} - \mathbf{A}\|_2 \leq \delta = \epsilon\,(\sigma_k(\mathbf{A}) - \sigma_{k+1}(\mathbf{A}))$ for some $\epsilon \in (0, 1/4]$. Let $\mathbf{A}_k$ and $\widehat{\mathbf{A}}_k$ be the best rank-$k$ approximations of $\mathbf{A}$ and $\widehat{\mathbf{A}}$. Then*

$$
\|\widehat{\mathbf{A}}_k - \mathbf{A}\|_F \leq \|\mathbf{A} - \mathbf{A}_k\|_F + 102\sqrt{2k}\epsilon\,(\sigma_k(\mathbf{A}) - \sigma_{k+1}(\mathbf{A})).
\tag{4}
$$

If $\mathbf{A}$ is an exact rank-$k$ matrix, Theorem 2.2 implies that truncated SVD gives an $\epsilon\sqrt{2k}\sigma_k$ error approximation in Frobenius norm, which has been established by many previous works [Yi et al., 2016, Tu et al., 2015, Wang et al., 2016].

Before we proceed to the applications and proof of Theorem 2.1, we first list several examples of $\mathbf{A}$ with classical distribution of eigenvalues and discuss how Theorem 2.1 could be applied to obatin good Frobenius-norm approximations of $\mathbf{A}$. We begin with the case where eigenvalues of $\mathbf{A}$ have a polynomial decay rate (i.e., power law). Such matrices are ubiquitous in practice [Liu et al., 2015].

**Corollary 2.1** (Power-law spectral decay). *Suppose $\|\hat{\mathbf{A}} - \mathbf{A}\|_2 \leq \delta$ for some $\delta \in (0, 1/2]$ and $\sigma_j(\mathbf{A}) = j^{-\beta}$ for some $\beta > 1/2$. Set $k = \lfloor \min\{C_1\delta^{-1/\beta}, n\} - 1 \rfloor$. If $k \geq 1$ then*

$$
\|\widehat{\mathbf{A}}_k - \mathbf{A}\|_F \leq C_1' \cdot \max\left\{\delta^{\frac{2\beta-1}{2\beta}}, n^{-\frac{2\beta-1}{2\beta}}\right\},
$$

*where $C_1, C_1' > 0$ are constants that only depend on $\beta$.*

We remark that the assumption $\sigma_j(\mathbf{A}) = j^{-\beta}$ implies that the eigenvalues lie in an $\ell_q$ ball for $q = 1/\beta$; that is, $\sum_{j=1}^n \sigma_j(\mathbf{A})^q = O(1)$. The error bound in Corollary 2.1 matches the minimax rate (derived by Negahban and Wainwright [2012]) for matrix completion when the spectrum is constrained in an $\ell_q$ ball, by replacing $\delta$ with $\sqrt{n/N}$ where $N$ is the number of observed entries.

Next, we consider the case where eigenvalues satisfy a faster decay rate.

**Corollary 2.2** (Exponential spectral decay). *Suppose $\|\hat{\mathbf{A}} - \mathbf{A}\|_2 \leq \delta$ for some $\delta \in (0, e^{-16})$ and $\sigma_j(\mathbf{A}) = \exp\{-cj\}$ for some $c > 0$. Set $k = \lfloor \min\{c^{-1}\log(1/\delta) - c^{-1}\log\log(1/\delta), n\} - 1 \rfloor$. If $k \geq 1$ then*

$$
\|\widehat{\mathbf{A}}_k - \mathbf{A}\|_F \leq C_2' \cdot \max\left\{\delta\sqrt{\log(1/\delta)^3}, n^{1/2}\exp(-cn)\right\},
$$

*where $C_2' > 0$ is a constant that only depends on $c$.*

Both corollaries are proved in the appendix. The error bounds in both Corollaries 2.1 and 2.2 are significantly better than the trivial estimate $\widehat{\mathbf{A}}$, which satisfies $\|\widehat{\mathbf{A}} - \mathbf{A}\|_F \leq n^{1/2}\delta$. We also remark that the bound in Corollary 2.1 cannot be obtained by a direct application of the weaker bound Eq. (3), which yields a $\delta^{\frac{\beta}{2\beta-1}}$ bound.

We next state results that are consequences of Theorem 2.1 in several matrix estimation problems.

## 2.1 High-rank Matrix Completion

Suppose $\mathbf{A}$ is a high-rank $n \times n$ PSD matrix that satisfies $\mu_0$-spikeness condition defined as follows:

**Definition 2.1** (Spikeness condition). *An $n \times n$ PSD matrix $\mathbf{A}$ satisfies $\mu_0$-spikeness condition if $n\|\mathbf{A}\|_{\max} \leq \mu_0\|\mathbf{A}\|_F$, where $\|\mathbf{A}\|_{\max} = \max_{1 \leq i,j \leq n} |\mathbf{A}_{ij}|$ is the max-norm of $\mathbf{A}$.*

Spikeness condition makes uniform sampling of matrix entries powerful in matrix completion problems. If $\mathbf{A}$ is exactly low rank, the spikeness condition is implied by an upper bound on $\max_{1 \leq i \leq n} \|\mathbf{e}_i^\top \mathbf{U}_k\|_2$, which is the standard incoherence assumption on the top-$k$ space of $\mathbf{A}$ [Candes and Recht, 2012]. For general high-rank $\mathbf{A}$, the spikeness condition is implied by a more restrictive incoherence condition that imposes an upper bound on $\max_{1 \leq i \leq n} \|\mathbf{e}_i^\top \mathbf{U}_{n-k}\|_2$ and $\|\mathbf{A}_{n-k}\|_{\max}$, which are assumptions adopted in [Hardt and Wootters, 2014].

Suppose $\widehat{\mathbf{A}}$ is a symmetric re-scaled zero-filled matrix of observed entries. That is,

$$[\widehat{\mathbf{A}}]_{ij} = \begin{cases} \mathbf{A}_{ij}/p, & \text{with probability } p; \\ 0, & \text{with probability } 1 - p; \end{cases} \quad \forall 1 \leq i \leq j \leq n. \tag{5}$$

Here $p \in (0,1)$ is a parameter that controls the probability of observing a particular entry in $\mathbf{A}$, corresponding to a sample complexity of $O(n^2 p)$. Note that both $\widehat{\mathbf{A}}$ and $\mathbf{A}$ are symmetric so we only specify the upper triangle of $\widehat{\mathbf{A}}$. By a simple application of matrix Bernstein inequality [Mackey et al., 2014], one can show $\widehat{\mathbf{A}}$ is close to $\mathbf{A}$ in spectral norm when $\mathbf{A}$ satisfies $\mu_0$-spikeness. Here we cite a lemma from [Hardt, 2014] to formally establish this observation:

**Lemma 2.1** (Corollary of [Hardt, 2014], Lemma A.3). *Under the model of Eq. (5) and $\mu_0$-spikeness condition of $\mathbf{A}$, for $t \in (0,1)$ it holds with probability at least $1 - t$ that*

$$\|\widehat{\mathbf{A}} - \mathbf{A}\|_2 \leq O\left(\max\left\{\sqrt{\frac{\mu_0^2\|\mathbf{A}\|_F^2 \log(n/t)}{np}}, \frac{\mu_0\|\mathbf{A}\|_F \log(n/t)}{np}\right\}\right).$$

Let $\widehat{\mathbf{A}}_k$ be the best rank-$k$ approximation of $\widehat{\mathbf{A}}$ in Frobenius/spectral norm. Applying Theorem 2.1 and 2.2 we obatin the following result:

**Theorem 2.3.** *Fix $t \in (0,1)$. Then with probability $1 - t$ we have*

$$\|\widehat{\mathbf{A}}_k - \mathbf{A}\|_F \leq O(\sqrt{k}) \cdot \|\mathbf{A} - \mathbf{A}_k\|_F \quad \text{if} \quad p = \Omega\left(\frac{\mu_0^2\|\mathbf{A}\|_F^2 \log(n/t)}{n\sigma_{k+1}(\mathbf{A})^2}\right).$$

*Furthermore, for fixed $\epsilon \in (0, 1/4]$, with probability $1 - t$ we have*

$$\|\widehat{\mathbf{A}}_k - \mathbf{A}\|_F \leq (1 + O(\epsilon))\|\mathbf{A} - \mathbf{A}_k\|_F \quad \text{if} \quad p = \Omega\left(\frac{\mu_0^2 \max\{\epsilon^{-4}, k^2\}\|\mathbf{A}\|_F^2 \log(n/t)}{n\sigma_{k+1}(\mathbf{A})^2}\right)$$

$$\|\widehat{\mathbf{A}}_k - \mathbf{A}\|_F \leq \|\mathbf{A} - \mathbf{A}_k\|_F + \epsilon\left(\sigma_k(\mathbf{A}) - \sigma_{k+1}(\mathbf{A})\right) \quad \text{if} \quad p = \Omega\left(\frac{\mu_0^2 k\|\mathbf{A}\|_F^2 \log(n/t)}{n\epsilon^2\left(\sigma_k(\mathbf{A}) - \sigma_{k+1}(\mathbf{A})\right)^2}\right).$$

As a remark, because $\mu_0 \geq 1$ and $\|\mathbf{A}\|_F/\sigma_{k+1}(\mathbf{A}) \geq \sqrt{k}$ *always* hold, the sample complexity is lower bounded by $\Omega(nk \log n)$, the typical sample complexity in noiseless matrix completion. In the case of high rank $\mathbf{A}$, the results in Theorem 2.3 are the strongest when $\mathbf{A}$ has small *stable rank* $r_s(\mathbf{A}) = \|\mathbf{A}\|_F^2/\|\mathbf{A}\|_2^2$ and the top-$k$ condition number $\gamma_k(\mathbf{A}) = \sigma_1(\mathbf{A})/\sigma_{k+1}(\mathbf{A})$ is not too large. For example, if $\mathbf{A}$ has stable rank $r_s(\mathbf{A}) = r$ then $\|\widehat{\mathbf{A}}_k - \mathbf{A}\|_F$ has an $O(\sqrt{k})$ multiplicative error bound with sample complexity $\Omega(\mu_0^2\gamma_k(\mathbf{A})^2 \cdot nr \log n)$; or an $(1 + O(\epsilon))$ relative error bound with sample complexity $\Omega(\mu_0^2 \max\{\epsilon^{-4}, k^2\}\gamma_k(\mathbf{A})^2 \cdot nr \log n)$. Finally, when $\sigma_{k+1}(\mathbf{A})$ is very small and the "gap" $\sigma_k(\mathbf{A}) - \sigma_{k+1}(\mathbf{A})$ is large, a weaker additive-error bound is applicable with sample complexity independent of $\sigma_{k+1}(\mathbf{A})^{-1}$.

Comparing with previous works, if' the gap $(1 - \sigma_{k+1}/\sigma_k)$ is of order $\epsilon$, then sample complexity of[Hardt, 2014] Theorem 1.2 and [Hardt and Wootters, 2014] Theorem 1 scale with $1/\epsilon^7$. Our result improves their results to the scaling of $1/\epsilon^4$ with a much simpler algorithm (truncated SVD).

## 2.2 High-rank matrix de-noising

Let $\mathbf{A}$ be an $n \times n$ PSD signal matrix and $\mathbf{E}$ a symmetric random Gaussian matrix with zero mean and $\nu^2/n$ variance. That is, $\mathbf{E}_{ij} \overset{i.i.d.}{\sim} \mathcal{N}(0, \nu^2/n)$ for $1 \leq i \leq j \leq n$ and $\mathbf{E}_{ij} = \mathbf{E}_{ji}$. Define $\widehat{\mathbf{A}} = \mathbf{A} + \mathbf{E}$. The matrix de-noising problem is then to recover the signal matrix $\mathbf{A}$ from noisy observations $\widehat{\mathbf{A}}$. We refer the readers to [Gavish and Donoho, 2014] for a list of references that shows the ubiquitous application of matrix de-noising in scientific fields.

It is well-known by concentration results of Gaussian random matrices, that $\|\widehat{\mathbf{A}} - \mathbf{A}\|_2 = \|\mathbf{E}\|_2 = O_{\mathbb{P}}(\nu)$. Let $\widehat{\mathbf{A}}_k$ be the best rank-$k$ approximation of $\widehat{\mathbf{A}}$ in Frobenius/spectral norm. Applying Theorem 2.1, we immediately have the following result:

**Theorem 2.4.** *There exists an absolute constant $c > 0$ such that, if $\nu < c \cdot \sigma_{k+1}(\mathbf{A})$ for some $1 \leq k < n$, then with probability at least 0.8 we have that*

$$\|\widehat{\mathbf{A}}_k - \mathbf{A}\|_F \leq \left(1 + O\left(\sqrt{\frac{\nu}{\sigma_{k+1}(\mathbf{A})}}\right)\right) \|\mathbf{A} - \mathbf{A}_k\|_F + O(\sqrt{k}\nu). \tag{6}$$

Eq. (6) can be understood from a classical bias-variance tradeoff perspective: the first $(1 + O(\sqrt{\nu/\sigma_{k+1}(\mathbf{A})}))\|\mathbf{A} - \mathbf{A}_k\|_F$ acts as a bias term, which decreases as we increase cut-off rank $k$, corresponding to a more complicated model; on the other hand, the second $O(\sqrt{k}\nu)$ term acts as the (square root of) variance, which does not depend on the signal $\mathbf{A}$ and increases with $k$.

## 2.3 Low-rank estimation of high-dimensional covariance

Suppose $\mathbf{A}$ is an $n \times n$ PSD matrix and $X_1, \cdots, X_N$ are i.i.d. samples drawn from the multivariate Gaussian distribution $\mathcal{N}_n(\mathbf{0}, \mathbf{A})$. The question is to estimate $\mathbf{A}$ from samples $X_1, \cdots, X_N$. A common estimator is the *sample covariance* $\widehat{\mathbf{A}} = \frac{1}{N}\sum_{i=1}^{N} X_i X_i^\top$. While in low-dimensional regimes (i.e., $n$ fixed and $N \to \infty$) the asymptotic efficiency of $\widehat{\mathbf{A}}$ is obvious (cf. [Van der Vaart, 2000]), its statistical power in high-dimensional regimes where $n$ and $N$ are comparable are highly non-trivial. Below we cite results by Bunea and Xiao [2015] for estimation error $\|\widehat{\mathbf{A}} - \mathbf{A}\|_\xi$, $\xi = 2/F$ when $n$ is not too large compared to $N$:

**Lemma 2.2** (Bunea and Xiao [2015]). *Suppose $n = O(N^\beta)$ for some $\beta \geq 0$ and let $r_e(\mathbf{A}) = \mathrm{tr}(\mathbf{A})/\|\mathbf{A}\|_2$ denote the* effective rank *of the covariance $\mathbf{A}$. Then the sample covariance $\widehat{\mathbf{A}} = \frac{1}{N}\sum_{i=1}^{N} X_i X_i^\top$ satisfies*

$$\|\widehat{\mathbf{A}} - \mathbf{A}\|_F = O_{\mathbb{P}}\left(\|\mathbf{A}\|_2 r_e(\mathbf{A})\sqrt{\frac{\log N}{N}}\right) \tag{7}$$

*and*

$$\|\widehat{\mathbf{A}} - \mathbf{A}\|_2 = O_{\mathbb{P}}\left(\|\mathbf{A}\|_2 \max\left\{\sqrt{\frac{r_e(\mathbf{A})\log(Nn)}{N}}, \frac{r_e(\mathbf{A})\log(Nn)}{N}\right\}\right). \tag{8}$$

Let $\widehat{\mathbf{A}}_k$ be the best rank-$k$ approximation of $\widehat{\mathbf{A}}$ in Frobenius/spectral norm. Applying Theorem 2.1 and 2.2 together with Eq. (8), we immediately arrive at the following theorem.

**Theorem 2.5.** *Fix $\epsilon \in (0, 1/4]$ and $1 \leq k < n$. Recall that $r_e(\mathbf{A}) = \mathrm{tr}(\mathbf{A})/\|\mathbf{A}\|_2$ and $\gamma_k(\mathbf{A}) = \sigma_1(\mathbf{A})/\sigma_{k+1}(\mathbf{A})$. There exists a universal constant $c > 0$ such that, if*

$$\frac{r_e(\mathbf{A})\max\{\epsilon^{-4}, k^2\}\gamma_k(\mathbf{A})^2 \log(N)}{N} \leq c$$

*then with probability at least 0.8,*

$$\|\widehat{\mathbf{A}}_k - \mathbf{A}\|_F \leq (1 + O(\epsilon))\|\mathbf{A} - \mathbf{A}_k\|_F$$

*and if*

$$\frac{r_e(\mathbf{A})k\|\mathbf{A}\|_2^2 \log(N)}{N\epsilon^2 (\sigma_k(\mathbf{A}) - \sigma_{k+1}(\mathbf{A}))^2} \leq c$$

*then with probability at least 0.8,*

$$\|\widehat{\mathbf{A}}_k - \mathbf{A}\|_F \leq \|\mathbf{A} - \mathbf{A}_k\|_F + \epsilon\left(\sigma_k\left(\mathbf{A}\right) - \sigma_{k+1}\left(\mathbf{A}\right)\right).$$

Theorem 2.5 shows that it is possible to obtain a reasonable Frobenius-norm approximation of $\widehat{\mathbf{A}}$ by truncated SVD in the asymptotic regime of $N = \Omega(r_e(\mathbf{A})\mathrm{poly}(k)\log N)$, which is much more flexible than Eq. (7) that requires $N = \Omega(r_e(\mathbf{A})^2 \log N)$.

## 3 Proof Sketch of Theorem 2.1

In this section we give a proof sketch of Theorem 2.1. The proof of Theorem 2.2 is similar and less challenging so we defer it to appendix. We defer proofs of technical lemmas to Section A.

Because both $\widehat{\mathbf{A}}_k$ and $\mathbf{A}_k$ are low-rank, $\|\widehat{\mathbf{A}}_k - \mathbf{A}_k\|_F$ is upper bounded by an $O(\sqrt{k})$ factor of $\|\widehat{\mathbf{A}}_k - \mathbf{A}_k\|_2$. From the condition that $\|\widehat{\mathbf{A}} - \mathbf{A}\|_2 \leq \delta$, a straightforward approach to upper bound $\|\widehat{\mathbf{A}}_k - \mathbf{A}_k\|_2$ is to consider the decomposition $\|\widehat{\mathbf{A}}_k - \mathbf{A}_k\|_2 \leq \|\widehat{\mathbf{A}} - \mathbf{A}\|_2 + 2\|\mathbf{U}_k\mathbf{U}_k^\top - \widehat{\mathbf{U}}_k\widehat{\mathbf{U}}_k^\top\|_2\|\widehat{\mathbf{A}}_k\|_2$, where $\mathbf{U}_k\mathbf{U}_k^\top$ and $\widehat{\mathbf{U}}_k\widehat{\mathbf{U}}_k^\top$ are projection operators onto the top-$k$ eigenspaces of $\mathbf{A}$ and $\widehat{\mathbf{A}}$, respectively. Such a naive approach, however, has two major disadvantages. First, the upper bound depends on $\|\widehat{\mathbf{A}}_k\|_2$, which is additive and may be much larger than $\|\widehat{\mathbf{A}} - \mathbf{A}\|_2$. Perhaps more importantly, the quantity $\|\mathbf{U}_k\mathbf{U}_k^\top - \widehat{\mathbf{U}}_k\widehat{\mathbf{U}}_k^\top\|_2$ depends on the "consecutive" spectral gap $(\sigma_k(\mathbf{A}) - \sigma_{k+1}(\mathbf{A}))$, which could be very small for large matrices.

The key idea in the proof of Theorem 2.1 is to find an "envelope" $m_1 \leq k \leq m_2$ in the spectrum of $\mathbf{A}$ surrounding $k$, such that the eigenvalues within the envelope are relatively close. Define

$$
\begin{aligned}
m_1 &= \mathrm{argmax}_{0 \leq j \leq k}\{\sigma_j(\mathbf{A}) \geq (1+2\epsilon)\sigma_{k+1}(\mathbf{A})\}; \\
m_2 &= \mathrm{argmax}_{k \leq j \leq n}\{\sigma_j(\mathbf{A}) \geq \sigma_k(\mathbf{A}) - 2\epsilon\sigma_{k+1}(\mathbf{A})\},
\end{aligned}
$$

where we let $\sigma_0(\mathbf{A}) = \infty$ for convenience. Let $\mathcal{U}_m, \widehat{\mathcal{U}}_m$ be basis of the top $m$-dimensional linear subspaces of $\mathbf{A}$ and $\widehat{\mathbf{A}}$, respectively. Also denote $\mathcal{U}_{n-m}$ and $\widehat{\mathcal{U}}_{n-m}$ as basis of the orthogonal complement of $\mathcal{U}_m$ and $\widehat{\mathcal{U}}_m$. By asymmetric Davis-Kahan inequality (Lemma C.1) and Wely's inequality we can obtain the following result.

**Lemma 3.1.** *If* $\|\widehat{\mathbf{A}} - \mathbf{A}\|_2 \leq \epsilon^2 \sigma_{k+1}(\mathbf{A})$ *for* $\epsilon \in (0,1)$ *then* $\|\widehat{\mathbf{U}}_{n-k}^\top \mathbf{U}_{m_1}\|_2, \|\widehat{\mathbf{U}}_k^\top \mathbf{U}_{n-m_2}\|_2 \leq \epsilon$.

Let $\mathcal{U}_{m_1:m_2}$ be the linear subspace of $\mathbf{A}$ associated with eigenvalues $\sigma_{m_1+1}(\mathbf{A}), \cdots, \sigma_{m_2}(\mathbf{A})$. Intuitively, we choose a $(k-m_1)$-dimensional linear subspace in $\mathcal{U}_{m_1:m_2}$ that is "most aligned" with the top-$k$ subspace $\widehat{\mathcal{U}}_k$ of $\widehat{\mathbf{A}}$. Formally, define

$$\mathcal{W} = \mathrm{argmax}_{\dim(\mathcal{W})=k-m_1, \mathcal{W} \in \mathcal{U}_{m_1:m_2}} \sigma_{k-m_1}\left(\mathbf{W}^\top \widehat{\mathbf{U}}_k\right).$$

$\mathbf{W}$ is then a $d \times (k-m_1)$ matrix with orthonormal columns that corresponds to a basis of $\mathcal{W}$. $\mathcal{W}$ is carefully constructed so that it is closely aligned with $\widehat{\mathcal{U}}_k$, yet still lies in $\mathcal{U}_k$. In particular, Lemma 3.2 shows that $\sin\angle(\mathcal{W}, \widehat{\mathcal{U}}_k) = \|\widehat{\mathbf{U}}_{n-k}^\top \mathbf{W}\|_2$ is upper bounded by $\epsilon$.

**Lemma 3.2.** *If* $\|\widehat{\mathbf{A}} - \mathbf{A}\|_2 \leq \epsilon^2 \sigma_{k+1}(\mathbf{A})$ *for* $\epsilon \in (0,1)$ *then* $\|\widehat{\mathbf{U}}_{n-k}^\top \mathbf{W}\|_2 \leq \epsilon$.

Now define

$$\widetilde{\mathbf{A}} = \mathbf{A}_{m_1} + \mathbf{W}\mathbf{W}^\top \mathbf{A}\mathbf{W}\mathbf{W}^\top.$$

We use $\widetilde{\mathbf{A}}$ as the "reference matrix" because we can decompose $\|\widehat{\mathbf{A}}_k - \mathbf{A}\|_F$ as

$$\|\widehat{\mathbf{A}}_k - \mathbf{A}\|_F \leq \|\mathbf{A} - \widetilde{\mathbf{A}}\|_F + \|\widehat{\mathbf{A}}_k - \widetilde{\mathbf{A}}\|_F \leq \|\mathbf{A} - \widetilde{\mathbf{A}}\|_F + \sqrt{2k}\|\widehat{\mathbf{A}}_k - \widetilde{\mathbf{A}}\|_2 \qquad (9)$$

and bound each term on the right hand side separately. Here the last inequality holds because both $\widehat{\mathbf{A}}_k$ and $\widetilde{\mathbf{A}}$ have rank at most $k$. The following lemma bounds the first term.

**Lemma 3.3.** *If* $\|\widehat{\mathbf{A}} - \mathbf{A}\|_2 \leq \epsilon^2 \sigma_{k+1}(\mathbf{A})^2$ *for* $\epsilon \in (0, 1/4]$ *then* $\|\mathbf{A} - \widetilde{\mathbf{A}}\|_F \leq (1+32\epsilon)\|\mathbf{A} - \mathbf{A}_k\|_F$.

The proof of this lemma relies Pythagorean theorem and Poincaré separation theorem. Let $\mathcal{U}_{m_1:m_2}$ be the $(m_2 - m_1)$-dimensional linear subspace such that $\mathcal{U}_{m_2} = \mathcal{U}_{m_1} \oplus \mathcal{U}_{m_1:m_2}$. Define $\mathbf{A}_{m_1:m_2} = \mathbf{U}_{m_1:m_2} \mathbf{\Sigma}_{m_1:m_2} \mathbf{U}_{m_1:m_2}^\top$, where $\mathbf{\Sigma}_{m_1:m_2} = \mathrm{diag}(\sigma_{m_1+1}(\mathbf{A}), \cdots, \sigma_{m_2}(\mathbf{A}))$ and $\mathbf{U}_{m_1:m_2}$ is an orthonormal basis associated with $\mathcal{U}_{m_1:m_2}$. Applying Pythagorean theorem (Lemma C.2), we can decompose

$$\|\mathbf{A} - \widetilde{\mathbf{A}}\|_F^2 = \|\mathbf{A} - \mathbf{A}_{m_2}\|_F^2 + \|\mathbf{A}_{m_1:m_2}\|_F^2 - \|\mathbf{W}\mathbf{W}^\top \mathbf{A}_{m_1:m_2} \mathbf{W}\mathbf{W}^\top\|_F^2.$$

Applying Poincaré separation theorem (Lemma C.3) where $\mathbf{X} = \mathbf{\Sigma}_{m_1:m_2}$ and $\mathbf{P} = \mathbf{U}_{m_1:m_2}^\top \mathbf{W}$, we have $\|\mathbf{W}^\top \mathbf{A}_{m_1:m_2} \mathbf{W}\|_F^2 \geq \sum_{j=m_2-k+1}^{m_2-m_1} \sigma_j(\mathbf{A}_{m_1:m_2})^2 = \sum_{j=m_1+m_2-k+1}^{m_2} \sigma_j(\mathbf{A})^2$. With some routine algebra we can prove Lemma 3.3.

To bound the second term of Eq. (9) we use the following lemma.

**Lemma 3.4.** *If $\|\widehat{\mathbf{A}} - \mathbf{A}\|_2 \leq \epsilon^2 \sigma_{k+1}(\mathbf{A})$ for $\epsilon \in (0, 1/4]$ then $\|\widehat{\mathbf{A}}_k - \widetilde{\mathbf{A}}\|_2 \leq 102\epsilon^2 \|\mathbf{A} - \mathbf{A}_k\|_2$.*

The proof of Lemma 3.4 relies on the low-rankness of $\widehat{\mathbf{A}}_k$ and $\widetilde{\mathbf{A}}$. Recall the definition that $\widetilde{\mathcal{U}} = \mathrm{Range}(\widetilde{\mathbf{A}})$ and $\widetilde{\mathcal{U}}_\perp = \mathrm{Null}(\widetilde{\mathbf{A}})$. Consider $\|\boldsymbol{v}\|_2 = 1$ such that $\boldsymbol{v}^\top (\widehat{\mathbf{A}}_k - \widetilde{\mathbf{A}})\boldsymbol{v} = \|\widehat{\mathbf{A}}_k - \widetilde{\mathbf{A}}\|_2$. Because $\boldsymbol{v}$ maximizes $\boldsymbol{v}^\top (\widehat{\mathbf{A}}_k - \widetilde{\mathbf{A}})\boldsymbol{v}$ over all unit-length vectors, it must lie in the range of $\left(\widehat{\mathbf{A}}_k - \widetilde{\mathbf{A}}\right)$ because otherwise the component outside the range will not contribute. Therefore, we can choose $\boldsymbol{v}$ that $\boldsymbol{v} = \boldsymbol{v}_1 + \boldsymbol{v}_2$ where $\boldsymbol{v}_1 \in \mathrm{Range}(\widehat{\mathbf{A}}_k) = \widehat{\mathcal{U}}_k$ and $\boldsymbol{v}_2 \in \mathrm{Range}(\widetilde{\mathbf{A}}) = \widetilde{\mathcal{U}}$. Subsequently, we have that

$$\begin{aligned} \boldsymbol{v} &= \widehat{\mathbf{U}}_k \widehat{\mathbf{U}}_k^\top \boldsymbol{v} + \widetilde{\mathbf{U}} \widetilde{\mathbf{U}}^\top \widehat{\mathbf{U}}_{n-k} \widehat{\mathbf{U}}_{n-k}^\top \boldsymbol{v} \qquad (10) \\ &= \widetilde{\mathbf{U}} \widetilde{\mathbf{U}}^\top \boldsymbol{v} + \widehat{\mathbf{U}}_k \widehat{\mathbf{U}}_k^\top \widetilde{\mathbf{U}}_\perp \widetilde{\mathbf{U}}_\perp^\top \boldsymbol{v}. \qquad (11) \end{aligned}$$

Consider the following decomposition:

$$\left|\boldsymbol{v}^\top (\widehat{\mathbf{A}}_k - \widetilde{\mathbf{A}})\boldsymbol{v}\right| \leq \left|\boldsymbol{v}^\top (\widehat{\mathbf{A}} - \mathbf{A})\boldsymbol{v}\right| + \left|\boldsymbol{v}^\top (\widehat{\mathbf{A}}_k - \widehat{\mathbf{A}})\boldsymbol{v}\right| + \left|\boldsymbol{v}^\top (\mathbf{A} - \widetilde{\mathbf{A}})\boldsymbol{v}\right|.$$

The first term $|\boldsymbol{v}^\top (\widehat{\mathbf{A}} - \mathbf{A})\boldsymbol{v}|$ is trivially upper bounded by $\|\widehat{\mathbf{A}} - \mathbf{A}\|_2 \leq \epsilon^2 \sigma_{k+1}(\mathbf{A})$. The second and the third term can be bounded by Wely's inequality (Lemma C.4) and basic properties of $\widetilde{\mathbf{A}}$ (Lemma A.3). See Section A for details.

## 4 Discussion

We mention two potential directions to further extend results of this paper.

### 4.1 Model selection for general high-rank matrices

The validity of Theorem 2.1 depends on the condition $\|\widehat{\mathbf{A}} - \mathbf{A}\|_2 \leq \epsilon^2 \sigma_{k+1}(\mathbf{A})$, which could be hard to verify if $\sigma_{k+1}(\mathbf{A})$ is unknown and difficult to estimate. Furthermore, for general high-rank matrices, the *model selection* problem of determining an appropriate (or even optimal) cut-off rank $k$ requires knowledge of the distribution of the entire spectrum of an unknown data matrix, which is even more challenging to obtain.

One potential approach is to impose a parametric pattern of decay of the eigenvalues (e.g., polynomial and exponential decay), and to estimate a small set of parameters (e.g., degree of polynomial) from the noisy observations $\widehat{\mathbf{A}}$. Afterwards, the optimal cut-off rank $k$ could be determined by a theoretical analysis, similar to the examples in Corollaries 2.1 and 2.2. Another possibility is to use repeated sampling techniques such as boostrap in a stochastic problem (e.g., matrix de-noising) to estimate the "bias" term $\|\mathbf{A} - \mathbf{A}_k\|_F$ for different $k$, as the variance term $\sqrt{k}\nu$ is known or easy to estimate.

### 4.2 Minimax rates for polynomial spectral decay

Consider the class of PSD matrices whose eigenvalues follow a polynomial (power-law) decay: $\Theta(\beta, n) = \{\mathbf{A} \in \mathbb{R}^{n \times n} : \mathbf{A} \succ 0, \sigma_j(\mathbf{A}) = j^{-\beta}\}$. We are interested in the following minimax rates for completing or de-noising matrices in $\Theta(\beta, n)$:

**Question 1** (Completion of $\Theta(\beta, n)$). *Fix $n \in \mathbb{N}$, $p \in (0, 1)$ and define $N = pn^2$. For $\mathbf{M} \in \Theta(\beta, n)$, let $\widehat{\mathbf{A}}_{ij} = \mathbf{M}_{ij}$ with probability $p$ and $\widehat{\mathbf{A}}_{ij} = 0$ with probability $1 - p$. Also let $\Lambda(\mu_0, n) = \{\mathbf{M} \in \mathbb{R}^{n \times n} : n\|\mathbf{M}\|_{\max} \leq \mu_0 \|\mathbf{M}\|_F\}$ be the class of all non-spiky matrices. Determine*

$$R_1(\mu_0, \beta, n, N) := \inf_{\widehat{\mathbf{A}} \mapsto \widehat{\mathbf{M}}} \sup_{\mathbf{M} \in \Theta(\beta, n) \cap \Lambda(\mu_0, n)} \mathbb{E}\|\widehat{\mathbf{M}} - \mathbf{M}\|_F^2.$$

**Question 2** (De-noising of $\Theta(\beta, n)$). *Fix $n \in \mathbb{N}$, $\nu > 0$ and let $\widehat{\mathbf{A}} = \mathbf{M} + \nu/\sqrt{n}\mathbf{Z}$, where $\mathbf{Z}$ is a symmetric matrices with i.i.d. standard Normal random variables on its upper triangle. Determine*

$$R_2(\nu, \beta, n) := \inf_{\widehat{\mathbf{A}} \mapsto \widehat{\mathbf{M}}} \sup_{\mathbf{M} \in \Theta(\beta, n)} \mathbb{E}\|\widehat{\mathbf{M}} - \mathbf{M}\|_F^2.$$

Compared to existing settings on matrix completion and de-noising, we believe $\Theta(\beta, n)$ is a more natural matrix class which allows for general high-rank matrices, but also imposes sufficient spectral decay conditions so that spectrum truncation algorithms result in significant benefits. Based on Corollary 2.1 and its matching lower bounds for a larger $\ell_p$ class [Negahban and Wainwright, 2012], we make the following conjecture:

**Conjecture 4.1.** *For $\beta > 1/2$ and $\nu$ not too small, we conjecture that*

$$R_1(\mu_0, \beta, n, N) \asymp C(\mu_0) \cdot \left[\frac{n}{N}\right]^{\frac{2\beta-1}{2\beta}} \quad and \quad R_2(\nu, \beta, n) \asymp \left[\nu^2\right]^{\frac{2\beta-1}{2\beta}},$$

*where $C(\mu_0) > 0$ is a constant that depends only on $\mu_0$.*

# 5  Acknowledgements

S.S.D. was supported by ARPA-E Terra program. Y.W. and A.S. were supported by the NSF CAREER grant IIS-1252412.

## Footnotes

[1] $n\|\mathbf{A}\|_{\max} \le \mu_0\|\mathbf{A}\|_F$; see also Definition 2.1.

[2] We remark that our relative-error analysis does *not*, however, apply to exact rank-$k$ matrix where $\sigma_{k+1} = 0$. This is because for exact rank-$k$ matrix a bound of the form $(1 + O(\epsilon))\|\mathbf{A} - \mathbf{A}_k\|_F$ requires *exact recovery* of $\mathbf{A}$, which truncated SVD cannot achieve. On the other hand, in the case of $\sigma_{k+1} = 0$ a weaker additive-error bound is always applicable, as we show in Theorem 2.3.

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
