[Supplementary Material]

# A  Proofs of Theorem 2.1 and Theorem 2.2

The key idea in the proof of Theorem 2.1 is to find an "envelope" $m_1 \le k \le m_2$ in the spectrum of $\mathbf{A}$ surrounding $k$, such that the eigenvalues within the envelope are relatively close. Define

$$
\begin{aligned}
m_1 &= \operatorname{argmax}_{0 \le j \le k} \{\sigma_j(\mathbf{A}) \ge (1+2\epsilon)\sigma_{k+1}(\mathbf{A})\}; \\
m_2 &= \operatorname{argmax}_{k \le j \le n} \{\sigma_j(\mathbf{A}) \ge \sigma_k(\mathbf{A}) - 2\epsilon\sigma_{k+1}(\mathbf{A})\},
\end{aligned}
$$

where we let $\sigma_0(\mathbf{A}) = \infty$ for convenience. Let $\mathcal{U}_m, \widehat{\mathcal{U}}_m$ be basis of the top $m$-dimensional linear subspaces of $\mathbf{A}$ and $\widehat{\mathbf{A}}$, respectively. Also denote $\mathcal{U}_{n-m}$ and $\widehat{\mathcal{U}}_{n-m}$ as basis of the orthogonal complement of $\mathcal{U}_m$ and $\widehat{\mathcal{U}}_m$.

**Lemma A.1.** *If* $\|\widehat{\mathbf{A}} - \mathbf{A}\|_2 \le \epsilon^2 \sigma_{k+1}(\mathbf{A})$ *for* $\epsilon \in (0,1)$ *then* $\|\widehat{\mathbf{U}}_{n-k}^\top \mathbf{U}_{m_1}\|_2, \|\widehat{\mathbf{U}}_k^\top \mathbf{U}_{n-m_2}\|_2 \le \epsilon$.

*Proof.* We apply an asymmetric version of Davis-Kahan inequality (Lemma C.1), with $\mathbf{X} = \mathbf{A}$, $\mathbf{Y} = \widehat{\mathbf{A}}$, $i = m_1$ and $j = k$. By Weyl's inequality, we know that $\sigma_{k+1}(\widehat{\mathbf{A}}) \le \sigma_{k+1}(\mathbf{A}) + \|\widehat{\mathbf{A}} - \mathbf{A}\|_2 \le (1+\epsilon^2)\sigma_{k+1}(\mathbf{A}) \le (1+\epsilon)\sigma_{k+1}(\mathbf{A})$. Subsequently, $\|\widehat{\mathbf{U}}_{n-k}^\top \mathbf{U}_{m_1}\|_2 \le \frac{\epsilon^2 \sigma_{k+1}(\mathbf{A})}{\sigma_{m_1}(\mathbf{A}) - (1+\epsilon)\sigma_{k+1}(\mathbf{A})} \le \epsilon$. Similarly, applying Lemma C.1 with $\mathbf{X} = \widehat{\mathbf{A}}$, $\mathbf{Y} = \mathbf{A}$, $i = k$ and $j = m_2$ we have that $\|\widehat{\mathbf{U}}_k^\top \mathbf{U}_{n-m_2}\|_2 \le \epsilon$. $\square$

Let $\mathcal{U}_{m_1:m_2}$ be the linear subspace of $\mathbf{A}$ associated with eigenvalues $\sigma_{m_1+1}(\mathbf{A}), \cdots, \sigma_{m_2}(\mathbf{A})$. Intuitively, we choose a $(k - m_1)$-dimensional linear subspace in $\mathcal{U}_{m_1:m_2}$ that is "most aligned" with the top-$k$ subspace $\widehat{\mathcal{U}}_k$ of $\widehat{\mathbf{A}}$. Formally, define

$$
\mathcal{W} = \operatorname{argmax}_{\dim(\mathcal{W}) = k - m_1, \mathcal{W} \in \mathcal{U}_{m_1:m_2}} \sigma_{k-m_1}\left(\mathbf{W}^\top \widehat{\mathbf{U}}_k\right).
$$

$\mathbf{W}$ is then a $d \times (k - m_1)$ matrix with orthonormal columns that corresponds to a basis of $\mathcal{W}$. $\mathcal{W}$ is carefully constructed so that it is closely aligned with $\widehat{\mathcal{U}}_k$, yet still lies in $\mathcal{U}_k$. In particular, Lemma 3.2 shows that $\sin\angle(\mathcal{W}, \widehat{\mathcal{U}}_k) = \|\widehat{\mathbf{U}}_{n-k}^\top \mathbf{W}\|_2$ is upper bounded by $\epsilon$.

**Lemma A.2.** *If* $\|\widehat{\mathbf{A}} - \mathbf{A}\|_2 \le \epsilon^2 \sigma_{k+1}(\mathbf{A})$ *for* $\epsilon \in (0,1)$ *then* $\|\widehat{\mathbf{U}}_{n-k}^\top \mathbf{W}\|_2 \le \epsilon$.

*Proof.* First note that $\|\widehat{\mathbf{U}}_{n-k}^\top \mathbf{W}\|_2 \le \sqrt{1 - \sigma_{k-m_1}(\widehat{\mathbf{U}}_k^\top \mathbf{W})^2}$ because

$$
\begin{aligned}
\|\widehat{\mathbf{U}}_{n-k}^\top \mathbf{W}\|_2^2 &= \sup_{\|\boldsymbol{x}\|_2 = 1} \|\widehat{\mathbf{U}}_{n-k}^\top \mathbf{W}\boldsymbol{x}\|_2^2 = \sup_{\|\boldsymbol{x}\|_2 = 1} \left\{\|\mathbf{W}\boldsymbol{x}\|_2^2 - \|\widehat{\mathbf{U}}_k^\top \mathbf{W}\boldsymbol{x}\|_2^2\right\} \\
&\le \sup_{\|\boldsymbol{x}\|_2 = 1} \|\mathbf{W}\boldsymbol{x}\|_2^2 - \inf_{\|\boldsymbol{x}\|_2 = 1} \|\widehat{\mathbf{U}}_k^\top \mathbf{W}\boldsymbol{x}\|_2^2 = 1 - \sigma_{k-m_1}(\widehat{\mathbf{U}}_k^\top \mathbf{W})^2.
\end{aligned}
$$

Subsequently, it suffices to prove that $\sigma_{k-m_1}(\widehat{\mathbf{U}}_k^\top \mathbf{W}) \ge \sqrt{1 - \epsilon^2}$. By Weyl's monotonicity theorem (Lemma C.4), we have that

$$
\sigma_k(\widehat{\mathbf{U}}_k^\top \mathbf{U}_{m_2}) \le \sigma_{m_1+1}(\widehat{\mathbf{U}}_k^\top \mathbf{U}_{m_1}) + \sigma_{k-m_1}(\widehat{\mathbf{U}}_k^\top \mathbf{U}_{m_1:m_2}).
$$

In addition, $\sigma_{m_1+1}(\widehat{\mathbf{U}}_k^\top \mathbf{U}_{m_1}) = 0$ because $\operatorname{rank}(\widehat{\mathbf{U}}_k^\top \mathbf{U}_{m_1}) \le m_1$ and $\sigma_{k-m_1}(\widehat{\mathbf{U}}_k^\top \mathbf{U}_{m_1:m_2}) = \sigma_{k-m_1}(\widehat{\mathbf{U}}_k^\top \mathbf{W})$ because of the definition of $\mathbf{W}$. Subsequently,

$$
\begin{aligned}
\sigma_{k-m_1}(\widehat{\mathbf{U}}_k^\top \mathbf{W})^2 &\ge \sigma_k(\widehat{\mathbf{U}}_k^\top \mathbf{U}_{m_2})^2 = \inf_{\|\boldsymbol{x}\|_2 = 1} \|\mathbf{U}_{m_2}^\top \widehat{\mathbf{U}}_k \boldsymbol{x}\|_2^2 = \inf_{\|\boldsymbol{x}\|_2 = 1} \left\{\|\widehat{\mathbf{U}}_k \vec{x}\|_2^2 - \|\mathbf{U}_{n-m_2}^\top \widehat{\mathbf{U}}_k \boldsymbol{x}\|_2^2\right\} \\
&\ge \inf_{\|\boldsymbol{x}\|_2 = 1} \left\{\|\widehat{\mathbf{U}}_k \boldsymbol{x}\|_2^2\right\} - \sup_{\|\boldsymbol{x}\|_2 = 1} \left\{\|\mathbf{U}_{n-m_2}^\top \widehat{\mathbf{U}}_k \boldsymbol{x}\|_2^2\right\} \ge 1 - \epsilon^2.
\end{aligned}
$$

Here in the last inequality we invoke Lemma 3.1. The proof is then complete. $\square$

Define

$$
\widetilde{\mathbf{A}} = \mathbf{A}_{m_1} + \mathbf{W}\mathbf{W}^\top \mathbf{A}\mathbf{W}\mathbf{W}^\top.
$$

The following lemma lists some of the properties of $\widetilde{\mathbf{A}}$.

**Lemma A.3.** *It holds that*

1. $\dim(\mathrm{Range}(\widetilde{\mathbf{A}})) = k$ *and* $\dim(\mathrm{Range}(\mathbf{W})) = k - m_1$;

2. $\mathcal{U}_{m_1} \subseteq \mathrm{Range}(\widetilde{\mathbf{A}}) \subseteq \mathcal{U}_{m_2}$ *and* $\mathrm{Range}(\widetilde{\mathbf{A}} - \mathbf{A}_{m_1}) \subseteq \mathcal{U}_{m_1:m_2}$, *where* $\mathcal{U}_{m_2} = \mathcal{U}_{m_1} \oplus \mathcal{U}_{m_1:m_2}$.

3. $\|\widehat{\mathbf{U}}_k^\top \widetilde{\mathbf{U}}_\perp\|_2, \|\widetilde{\mathbf{U}}^\top \widehat{\mathbf{U}}_{n-k}\|_2 \leq 2\epsilon$, *where* $\widetilde{\mathbf{U}}$ *and* $\widetilde{\mathbf{U}}_\perp$ *are orthonormal basis of* $\mathrm{Range}(\widetilde{\mathbf{A}})$ *and* $\mathrm{Null}(\widetilde{\mathbf{A}})$, *respectively.*

*Proof.* Properties 1 and 2 are obviously true by the definition of $\mathcal{W}$ and $\widetilde{\mathbf{A}}$. For property 3, note that both $\|\widehat{\mathbf{U}}_k^\top \widetilde{\mathbf{U}}_\perp\|_2$ and $\|\widetilde{\mathbf{U}}^\top \widehat{\mathbf{U}}_{n-k}\|_2$ are equal to $\sin \angle(\widetilde{\mathcal{U}}, \widehat{\mathcal{U}}_k)$. Hence it suffices to show that $\|\widehat{\mathbf{U}}_{n-k}^\top \widetilde{\mathbf{U}}\|_2 \leq 2\epsilon$. Invoking Lemmas 3.1 and 3.2 we have that $\|\widehat{\mathbf{U}}_{n-k}^\top \widetilde{\mathbf{U}}\|_2 \leq \|\widehat{\mathbf{U}}_{n-k}^\top \mathbf{U}_{m_1}\|_2 + \|\widehat{\mathbf{U}}_{n-k}^\top \mathbf{W}\|_2 \leq \epsilon + \epsilon = 2\epsilon$. $\qquad\qquad\square$

Decompose $\|\widehat{\mathbf{A}}_k - \mathbf{A}\|_F$ as

$$\|\widehat{\mathbf{A}}_k - \mathbf{A}\|_F \leq \|\mathbf{A} - \widetilde{\mathbf{A}}\|_F + \|\widehat{\mathbf{A}}_k - \widetilde{\mathbf{A}}\|_F \leq \|\mathbf{A} - \widetilde{\mathbf{A}}\|_F + \sqrt{2k}\|\widehat{\mathbf{A}}_k - \widetilde{\mathbf{A}}\|_2. \qquad (12)$$

Here the last inequality holds because both $\widehat{\mathbf{A}}_k$ and $\widetilde{\mathbf{A}}$ have rank at most $k$. Lemmas 3.3 and 3.4 give separate upper bounds for $\|\mathbf{A} - \widetilde{\mathbf{A}}\|_F$ and $\|\widehat{\mathbf{A}}_k - \widetilde{\mathbf{A}}\|_2$.

**Lemma A.4.** *If* $\|\widehat{\mathbf{A}} - \mathbf{A}\|_2 \leq \epsilon^2 \sigma_{k+1}(\mathbf{A})^2$ *for* $\epsilon \in (0, 1/4]$ *then* $\|\mathbf{A} - \widetilde{\mathbf{A}}\|_F \leq (1 + 32\epsilon)\|\mathbf{A} - \mathbf{A}_k\|_F$.

*Proof.* Let $\mathcal{U}_{m_1:m_2}$ be the $(m_2 - m_1)$-dimensional linear subspace such that $\mathcal{U}_{m_2} = \mathcal{U}_{m_1} \oplus \mathcal{U}_{m_1:m_2}$. Define $\mathbf{A}_{m_1:m_2} = \mathbf{U}_{m_1:m_2}\boldsymbol{\Sigma}_{m_1:m_2}\mathbf{U}_{m_1:m_2}^\top$, where $\boldsymbol{\Sigma}_{m_1:m_2} = \mathrm{diag}(\sigma_{m_1+1}(\mathbf{A}), \cdots, \sigma_{m_2}(\mathbf{A}))$ and $\mathbf{U}_{m_1:m_2}$ is an orthonormal basis associated with $\mathcal{U}_{m_1:m_2}$. We then have

$$\begin{aligned}
\|\mathbf{A} - \widetilde{\mathbf{A}}\|_F^2 &= \|\mathbf{A}_{n-m_1} - \mathbf{W}\mathbf{W}^\top \mathbf{A}\mathbf{W}\mathbf{W}^\top\|_F^2 \\
&\overset{(a)}{=} \|\mathbf{A}_{n-m_2}\|_F^2 + \|\mathbf{A}_{m_1:m_2} - \mathbf{W}\mathbf{W}^\top \mathbf{A}\mathbf{W}\mathbf{W}^\top\|_F^2 \\
&\overset{(b)}{=} \|\mathbf{A} - \mathbf{A}_{m_2}\|_F^2 + \|\mathbf{A}_{m_1:m_2} - \mathbf{W}\mathbf{W}^\top \mathbf{A}_{m_1:m_2}\mathbf{W}\mathbf{W}^\top\|_F^2 \\
&\overset{(c)}{=} \|\mathbf{A} - \mathbf{A}_{m_2}\|_F^2 + \|\mathbf{A}_{m_1:m_2}\|_F^2 - \|\mathbf{W}\mathbf{W}^\top \mathbf{A}_{m_1:m_2}\mathbf{W}\mathbf{W}^\top\|_F^2.
\end{aligned}$$

Here in $(a)$ we apply $\mathrm{Range}(\widetilde{\mathbf{A}} - \mathbf{A}_{m_1}) \subseteq \mathcal{U}_{m_1:m_2}$ and the Pythagorean theorem (Lemma C.2) with $\mathbf{P} = \mathbf{U}_{m_1:m_2}$, in $(b)$ we apply $\mathcal{W} \subseteq \mathcal{U}_{m_1:m_2}$, and in $(c)$ we apply the Pythagorean theorem again with $\mathbf{P} = \mathbf{W}$. Note that $\|\mathbf{W}\mathbf{W}^\top \mathbf{A}_{m_1:m_2}\mathbf{W}\mathbf{W}^\top\|_F^2 = \|\mathbf{W}^\top \mathbf{A}_{m_1:m_2}\mathbf{W}\|_F^2$. Applying Poincaré separation theorem (Lemma C.3) where $\mathbf{X} = \boldsymbol{\Sigma}_{m_1:m_2}$ and $\mathbf{P} = \mathbf{U}_{m_1:m_2}^\top\mathbf{W}$, we have $\|\mathbf{W}^\top \mathbf{A}_{m_1:m_2}\mathbf{W}\|_F^2 \geq \sum_{j=m_2-k+1}^{m_2-m_1} \sigma_j(\mathbf{A}_{m_1:m_2})^2 = \sum_{j=m_1+m_2-k+1}^{m_2} \sigma_j(\mathbf{A})^2$. Subsequently,

$$\begin{aligned}
\|\mathbf{A} - \widetilde{\mathbf{A}}\|_F^2 &\leq \|\mathbf{A} - \mathbf{A}_{m_2}\|_F^2 + \sum_{j=m_1+1}^{m_1+m_2-k} \sigma_j(\mathbf{A})^2 \leq \|\mathbf{A} - \mathbf{A}_{m_2}\|_F^2 + (m_2 - k)\sigma_{m_1+1}(\mathbf{A})^2 \\
&\overset{(a')}{\leq} \|\mathbf{A} - \mathbf{A}_{m_2}\|_F^2 + (m_2 - k)(1 + 2\epsilon)^2 \sigma_{k+1}(\mathbf{A})^2 \\
&\overset{(b')}{\leq} \|\mathbf{A} - \mathbf{A}_{m_2}\|_F^2 + (m_2 - k)\left(\frac{1 + 2\epsilon}{1 - 2\epsilon}\right)^2 \sigma_{m_2}(\mathbf{A})^2 \\
&\overset{(c')}{\leq} \|\mathbf{A} - \mathbf{A}_{m_2}\|_F^2 + (m_2 - k)\sigma_{m_2}(\mathbf{A})^2 + 32(m_2 - k)\epsilon\sigma_{m_2}(\mathbf{A})^2 \\
&\overset{(d')}{\leq} (1 + 32\epsilon)\|\mathbf{A} - \mathbf{A}_k\|_F^2.
\end{aligned}$$

Here in $(a')$ we apply the definition of $m_1$ that $\sigma_{m_1+1} \leq (1 + 2\epsilon)\sigma_{k+1}(\mathbf{A})$, in $(b')$ we apply the definition of $m_2$ that $\sigma_{m_2}(\mathbf{A}) \geq \sigma_k(\mathbf{A}) - 2\epsilon\sigma_{k+1}(\mathbf{A}) \geq (1 - 2\epsilon)\sigma_{k+1}(\mathbf{A})$, and $(c')$ is due to the fact that $\left(\frac{1+2\epsilon}{1-2\epsilon}\right)^2 \leq 1 + 32\epsilon$ for all $\epsilon \in (0, 1/4]$. Finally, $(d')$ holds because $(m_2 - k)\sigma_{m_2}(\mathbf{A})^2 \leq \sum_{j=k+1}^{m_2} \sigma_j(\mathbf{A})^2$ and $\|\mathbf{A} - \mathbf{A}_k\|_F^2 = \|\mathbf{A} - \mathbf{A}_{m_2}\|_F^2 + \sum_{j=k+1}^{m_2} \sigma_j(\mathbf{A})^2$. $\qquad\square$

**Lemma A.5.** *If* $\|\widehat{\mathbf{A}} - \mathbf{A}\|_2 \le \epsilon^2 \sigma_{k+1}(\mathbf{A})$ *for* $\epsilon \in (0, 1/4]$ *then* $\|\widehat{\mathbf{A}}_k - \widetilde{\mathbf{A}}\|_2 \le 102\epsilon^2 \|\mathbf{A} - \mathbf{A}_k\|_2$.

*Proof.* Recall the definition that $\widetilde{\mathcal{U}} = \mathrm{Range}(\widetilde{\mathbf{A}})$ and $\widetilde{\mathcal{U}}_\perp = \mathrm{Null}(\widetilde{\mathbf{A}})$. Consider $\|\boldsymbol{v}\|_2 = 1$ such that $\boldsymbol{v}^\top(\widehat{\mathbf{A}}_k - \widetilde{\mathbf{A}})\boldsymbol{v} = \|\widehat{\mathbf{A}}_k - \widetilde{\mathbf{A}}\|_2$. Because $\boldsymbol{v}$ maximizes $\boldsymbol{v}^\top(\widehat{\mathbf{A}}_k - \widetilde{\mathbf{A}})\boldsymbol{v}$ over all unit-length vectors, it must lie in the range of $\left(\widehat{\mathbf{A}}_k - \widetilde{\mathbf{A}}\right)$ because otherwise the component outside the range will not contribute. Therefore, we can choose $\boldsymbol{v}$ that $\boldsymbol{v} = \boldsymbol{v}_1 + \boldsymbol{v}_2$ where $\boldsymbol{v}_1 \in \mathrm{Range}(\widehat{\mathbf{A}}_k) = \widehat{\mathcal{U}}_k$ and $\boldsymbol{v}_2 \in \mathrm{Range}(\widetilde{\mathbf{A}}) = \widetilde{\mathcal{U}}$. Subsequently, we have that

$$\boldsymbol{v} = \widehat{\mathbf{U}}_k \widehat{\mathbf{U}}_k^\top \boldsymbol{v} + \widetilde{\mathbf{U}}\widetilde{\mathbf{U}}^\top \widehat{\mathbf{U}}_{n-k}\widehat{\mathbf{U}}_{n-k}^\top \boldsymbol{v} \tag{13}$$

$$= \widetilde{\mathbf{U}}\widetilde{\mathbf{U}}^\top \boldsymbol{v} + \widehat{\mathbf{U}}_k \widehat{\mathbf{U}}_k^\top \widetilde{\mathbf{U}}_\perp \widetilde{\mathbf{U}}_\perp^\top \boldsymbol{v}. \tag{14}$$

Consider the following decomposition:

$$\left|\boldsymbol{v}^\top(\widehat{\mathbf{A}}_k - \widetilde{\mathbf{A}})\boldsymbol{v}\right| \le \left|\boldsymbol{v}^\top(\widehat{\mathbf{A}} - \mathbf{A})\boldsymbol{v}\right| + \left|\boldsymbol{v}^\top(\widehat{\mathbf{A}}_k - \widehat{\mathbf{A}})\boldsymbol{v}\right| + \left|\boldsymbol{v}^\top(\mathbf{A} - \widetilde{\mathbf{A}})\boldsymbol{v}\right|.$$

The first term $|\boldsymbol{v}^\top(\widehat{\mathbf{A}} - \mathbf{A})\boldsymbol{v}|$ is trivially upper bounded by $\|\widehat{\mathbf{A}} - \mathbf{A}\|_2 \le \epsilon^2 \sigma_{k+1}(\mathbf{A})$. For the second term, we have

$$\left|\boldsymbol{v}^\top(\widehat{\mathbf{A}}_k - \widehat{\mathbf{A}})\boldsymbol{v}\right| = \left|\boldsymbol{v}^\top \widehat{\mathbf{U}}_{n-k}\widehat{\boldsymbol{\Sigma}}_{n-k}\widehat{\mathbf{U}}_{n-k}^\top \boldsymbol{v}\right\|$$

$$\overset{(a)}{=} \left|\boldsymbol{v}^\top \widehat{\mathbf{U}}_{n-k}\widehat{\mathbf{U}}_{n-k}^\top \widetilde{\mathbf{U}}\widetilde{\mathbf{U}}^\top \widehat{\mathbf{U}}_{n-k}\widehat{\boldsymbol{\Sigma}}_{n-k}\widehat{\mathbf{U}}_{n-k}^\top \widetilde{\mathbf{U}}\widetilde{\mathbf{U}}^\top \widehat{\mathbf{U}}_{n-k}\widehat{\mathbf{U}}_{n-k}^\top \boldsymbol{v}\right|$$

$$\le \left\|\widehat{\mathbf{U}}_{n-k}^\top \widetilde{\mathbf{U}}\right\|_2^4 \left\|\widehat{\mathbf{U}}_{n-k}\right\|_2 \overset{(b)}{\le} 16\epsilon^4 \sigma_{k+1}(\widehat{\mathbf{A}}) \overset{(c)}{\le} 16\epsilon^4(1 + \epsilon^2)\sigma_{k+1}(\mathbf{A}).$$

Here in $(a)$ we apply Eq. (10); in $(b)$ we apply Property 3 of Lemma A.3, and $(c)$ is due to Weyl's inequality (Lemma C.4) that $\sigma_{k+1}(\widehat{\mathbf{A}}) \le \sigma_{k+1}(\mathbf{A}) + \|\widehat{\mathbf{A}} - \mathbf{A}\|_2 \le (1 + \epsilon^2)\sigma_{k+1}(\mathbf{A})$.

For the third term, note that $\widetilde{\mathbf{A}} = \widetilde{\mathbf{U}}\widetilde{\mathbf{U}}^\top \mathbf{A}\widetilde{\mathbf{U}}\widetilde{\mathbf{U}}^\top$ because $\mathrm{Range}(\widetilde{\mathbf{A}}) \subseteq \mathcal{U}_{m_2} \subseteq \mathrm{Range}(\mathbf{A})$ by Lemma A.3. Subsequently,

$$\mathbf{A} - \widetilde{\mathbf{A}} = \underbrace{\widetilde{\mathbf{U}}_\perp \widetilde{\mathbf{U}}_\perp^\top \mathbf{A}\widetilde{\mathbf{U}}_\perp \widetilde{\mathbf{U}}_\perp^\top}_{\mathbf{B}_1} + \underbrace{\widetilde{\mathbf{U}}\widetilde{\mathbf{U}}^\top \mathbf{A}\widetilde{\mathbf{U}}_\perp \widetilde{\mathbf{U}}_\perp^\top}_{\mathbf{B}_2} + \underbrace{\widetilde{\mathbf{U}}_\perp \widetilde{\mathbf{U}}_\perp^\top \mathbf{A}\widetilde{\mathbf{U}}\widetilde{\mathbf{U}}^\top}_{\mathbf{B}_2^\top}.$$

It then suffices to upper bound $|\boldsymbol{v}^\top \mathbf{B}_1 \boldsymbol{v}|$ and $|\boldsymbol{v}^\top \mathbf{B}_2 \boldsymbol{v}|$ separately. For $\mathbf{B}_1$ we have

$$\left|\boldsymbol{v}^\top \mathbf{B}_1 \boldsymbol{v}\right| \overset{(a')}{=} \left|\boldsymbol{v}^\top \widetilde{\mathbf{U}}_\perp \widetilde{\mathbf{U}}_\perp^\top \widehat{\mathbf{U}}_k \widehat{\mathbf{U}}_k^\top \widetilde{\mathbf{U}}_\perp \widetilde{\mathbf{U}}_\perp^\top \mathbf{A}\widetilde{\mathbf{U}}_\perp \widetilde{\mathbf{U}}_\perp^\top \widehat{\mathbf{U}}_k \widehat{\mathbf{U}}_k^\top \widetilde{\mathbf{U}}_\perp \widetilde{\mathbf{U}}_\perp^\top \boldsymbol{v}\right|$$

$$\le \left\|\widetilde{\mathbf{U}}_\perp^\top \widehat{\mathbf{U}}_k\right\|_2^4 \left\|\widetilde{\mathbf{U}}_\perp^\top \mathbf{A}\widetilde{\mathbf{U}}_\perp\right\|_2$$

$$\overset{(b')}{\le} 16\epsilon^4 \left\|\widetilde{\mathbf{U}}_\perp^\top \mathbf{A}\widetilde{\mathbf{U}}_\perp\right\|_2 \overset{(c')}{\le} 16\epsilon^4 \sigma_{m_1+1}(\mathbf{A}) \overset{(d')}{\le} 16\epsilon^4(1 + 2\epsilon)\sigma_{k+1}(\mathbf{A}).$$

Here in $(a')$ we apply Eq. (11); in $(b')$ we apply Property 3 of Lemma A.3; $(c')$ follows the property that $\widetilde{\mathcal{U}}_\perp \in \mathcal{U}_{n-m_1}$, and finally $(d')$ follows from the definition of $m_1$ that $\sigma_{m_1+1}(\mathbf{A}) \le (1 + 2\epsilon)\sigma_{k+1}(\mathbf{A})$.

For $\mathbf{B}_2$, we have that

$$\left|\boldsymbol{v}^\top \mathbf{B}_2 \boldsymbol{v}\right| = \left|\boldsymbol{v}^\top \widetilde{\mathbf{U}}\widetilde{\mathbf{U}}^\top \mathbf{A}\widetilde{\mathbf{U}}_\perp \widetilde{\mathbf{U}}_\perp^\top \widehat{\mathbf{U}}_k \widehat{\mathbf{U}}_k^\top \widetilde{\mathbf{U}}_\perp \widetilde{\mathbf{U}}_\perp^\top \boldsymbol{v}\right|$$

$$\le \left\|\mathbf{A}\widetilde{\mathbf{U}}_\perp\right\|_2 \left\|\widetilde{\mathbf{U}}_\perp^\top \widehat{\mathbf{U}}_k\right\|_2^2 \le \epsilon^2(1 + 8\epsilon)\sigma_{k+1}(\mathbf{A}).$$

Combining all inequalities and noting that $\epsilon \in (0, 1/4]$, we obtain

$$\|\widehat{\mathbf{A}}_k - \widetilde{\mathbf{A}}\|_2 \le \epsilon^2 \sigma_{k+1}(\mathbf{A}) + 16\epsilon^4(1 + 2\epsilon + \epsilon^2)\sigma_{k+1}(\mathbf{A}) + 32\epsilon^2(1 + 8\epsilon)\sigma_{k+1}(\mathbf{A})$$

$$\le 102\epsilon^2 \sigma_{k+1}(\mathbf{A}).$$

$\square$

*Proof.* **of Theorem 2.2** The proof of Theorem 2.2 is similar and even simpler than that of Theorem 2.1. First observing that with the large spectral gap, $\widetilde{\mathbf{A}} = \mathbf{A}_k$. Next we replace by replacing the assumption $\|\widehat{\mathbf{A}} - \mathbf{A}\|_2 \leq \epsilon^2 \sigma_{k+1}(\mathbf{A})$ in Lemma 3.4 with $\|\widehat{\mathbf{A}} - \mathbf{A}\|_2 \leq \epsilon (\sigma_k(\mathbf{A}) - \sigma_{k+1}(\mathbf{A}))$ using the exactly the same arguments we have

$$\|\widehat{\mathbf{A}}_k - \mathbf{A}_k\|_2 \leq 102\epsilon (\sigma_k(\mathbf{A}) - \sigma_{k+1}(\mathbf{A})).$$

Therefore, we have

$$\|\widehat{\mathbf{A}}_k - \mathbf{A}_k\|_F \leq 102\sqrt{2k}\epsilon (\sigma_k(\mathbf{A}) - \sigma_{k+1}(\mathbf{A})).$$

Lastly, apply triangle inequality:

$$\begin{aligned}
\|\widehat{\mathbf{A}}_k - \mathbf{A}\|_F &\leq \|\mathbf{A} - \mathbf{A}_k\|_F + \|\widehat{\mathbf{A}}_k - \mathbf{A}_k\|_F \\
&\leq \|\mathbf{A} - \mathbf{A}_k\|_F + 102\sqrt{2k}\epsilon (\sigma_k(\mathbf{A}) - \sigma_{k+1}(\mathbf{A})).
\end{aligned}$$

$\square$

## B    Proof of corollaries

*Proof.* **of Corollary 2.1**. We first verify the condition that $\delta \leq \epsilon^2 \sigma_{k+1}(\mathbf{A})$ for $\epsilon = 1/4$ and the particular choice of $k$. Because $k \leq \lfloor C_1 \delta^{-1/\beta} \rfloor - 1$, we have that $\sigma_{k+1}(\mathbf{A}) \geq (C_1 \delta^{-1/\beta})^{-\beta}$. By carefully chosen $C_1$ (depending on $\beta$) the inequality $\sigma_{k+1}(\mathbf{A}) \geq \delta/16$ holds.

If $k = n - 1$ then by Theorem 2.1, $\|\widehat{\mathbf{A}}_k - \mathbf{A}\|_F \leq O(\sqrt{n} \cdot n^{-\beta}) = O(n^{-\frac{2\beta-1}{2\beta}})$. In the rest of the proof we assume $k = \lfloor C_1 \delta^{-1/\beta} \rfloor - 1$. We then have

$$\|\mathbf{A} - \mathbf{A}_k\|_F = \sqrt{\sum_{j=k+1}^{n} \sigma_j(\mathbf{A})^2} = \sqrt{\sum_{j=k+1}^{n} j^{-2\beta}} \leq \sqrt{\int_k^\infty x^{-2\beta} dx} = \sqrt{\frac{k^{-(2\beta-1)}}{2\beta-1}} \leq C(\beta)\delta^{\frac{2\beta-1}{2\beta}}.$$

Here $C(\beta) > 0$ is a constant that only depends on $\beta$. In addition,

$$\sqrt{k}\|\mathbf{A} - \mathbf{A}_k\|_2 \leq \sqrt{k} \cdot k^{-\beta} = k^{-(\beta-1/2)} \leq \tilde{C}(\beta)\delta^{\frac{2\beta}{2\beta-1}}.$$

Applying Theorem 2.1 we complete the proof of Corollary 2.1. $\square$

*Proof.* **of Corollary 2.2** We first verify the condition that $\delta \leq \epsilon^2 \sigma_{k+1}(\mathbf{A})$ for $\epsilon = 1/4$ and the particular choice of $k$. Because $k \leq \lfloor c^{-1} \log(1/\delta) - c^{-1} \log\log(1/\delta) \rfloor - 1$, we have that $\sigma_{k+1}(\mathbf{A}) \geq \delta \log(1/\delta)$. Hence, for $\delta \in (0, e^{-16})$ it holds that $\sigma_{k+1}(\mathbf{A}) \geq \delta/16$.

If $k = n - 1$ then by Theorem 2.1, $\|\widehat{\mathbf{A}}_k - \mathbf{A}\|_F \leq O(\sqrt{n} \cdot \exp\{-cn\})$. In the rest of the proof we assume $k = \lfloor C_2 \log(1/\delta) \rfloor - 1$. We then have

$$\|\mathbf{A} - \mathbf{A}_k\|_F = \sqrt{\sum_{j=k+1}^{n} \sigma_j(\mathbf{A})^2} = \sqrt{\sum_{j=k+1}^{n} \exp\{-2cj\}} \leq \sqrt{\frac{\exp\{-2ck\}}{1 - e^{-2c}}} \leq C(c)\delta \log(1/\delta),$$

where $C(c) > 0$ is a constant that only depends on $c$. In addition,

$$\sqrt{k}\|\mathbf{A} - \mathbf{A}_k\|_2 \leq \sqrt{k} \cdot \exp\{-ck\} \leq \delta \log(1/\delta) \cdot \sqrt{c^{-1}\log(1/\delta)} \leq \tilde{C}(c)\delta\sqrt{\log(1/\delta)^3}.$$

Applying Theorem 2.1 we complete the proof of Corollary 2.2. $\square$

## C    Technical lemmas

**Lemma C.1** (Asymmetric Davis-Kahan inequality). *Fix $i \leq j \leq n$ and suppose $\mathbf{X}, \mathbf{Y}$ are symmetric $n \times n$ matrices, with eigen-decomposition $\mathbf{X} = \mathbf{P}_i \mathbf{\Lambda}_i \mathbf{P}_i^\top + \mathbf{P}_{n-i} \mathbf{\Lambda}_{n-i} \mathbf{P}_{n-i}^\top$ and $\mathbf{Y} = \mathbf{Q}_j \mathbf{\Xi}_j \mathbf{Q}_j^\top + \mathbf{Q}_{n-j} \mathbf{\Xi}_{n-j} \mathbf{Q}_{n-j}^\top$. If $\sigma_i(\mathbf{X}) > \sigma_{j+1}(\mathbf{Y})$ then*

$$\|\mathbf{Q}_{n-j}^\top \mathbf{P}_i\|_2 \leq \frac{\|\mathbf{X} - \mathbf{Y}\|_2}{\sigma_i(\mathbf{X}) - \sigma_{j+1}(\mathbf{Y})}.$$

*Proof.* Consider

$$\left\|\mathbf{Q}_{n-j}^\top(\mathbf{X}-\mathbf{Y})\mathbf{P}_i\right\|_2 = \left\|\mathbf{Q}_{n-j}^\top\mathbf{P}_i\mathbf{\Lambda}_i - \mathbf{\Xi}_{n-j}\mathbf{Q}_{n-j}^\top\mathbf{P}_i\right\|_2 \geq \left\|\mathbf{Q}_{n-j}^\top\mathbf{P}_i\right\|_2 (\sigma_i(\mathbf{X}) - \sigma_{j+1}(\mathbf{Y})).$$

Because $\sigma_i(\mathbf{X}) > \sigma_{j+1}(\mathbf{Y})$, we have that

$$\left\|\mathbf{Q}_{n-j}^\top\mathbf{P}_i\right\|_2 \leq \frac{\|\mathbf{Q}_{n-j}^\top(\mathbf{X}-\mathbf{Y})\mathbf{P}_i\|_2}{\sigma_i(\mathbf{X}) - \sigma_{j+1}(\mathbf{Y})} \leq \frac{\|\mathbf{X}-\mathbf{Y}\|_2}{\sigma_i(\mathbf{X}) - \sigma_{j+1}(\mathbf{Y})}.$$

$\square$

**Lemma C.2** (Pythagorean theorem). *Fix $n \geq m$. Suppose $\mathbf{X}$ is a symmetric $n \times n$ matrix and $\mathbf{P}$ is an $n \times m$ matrix satisfying $\mathbf{P}^\top\mathbf{P} = \mathbf{I}$. Then $\|\mathbf{X}\|_F^2 = \|\mathbf{X} - \mathbf{P}\mathbf{P}^\top\mathbf{X}\mathbf{P}\mathbf{P}^\top\|_F^2 + \|\mathbf{P}\mathbf{P}^\top\mathbf{X}\mathbf{P}\mathbf{P}^\top\|_F^2$.*

*Proof.* Expanding $\|\mathbf{X}\|_F^2$ we have that

$$\begin{aligned}
\|\mathbf{X}\|_F^2 &= \|(\mathbf{X} - \mathbf{P}\mathbf{P}^\top\mathbf{X}\mathbf{P}\mathbf{P}^\top) + \mathbf{P}\mathbf{P}^\top\mathbf{X}\mathbf{P}\mathbf{P}^\top\|_F^2 \\
&= \|\mathbf{X} - \mathbf{P}\mathbf{P}^\top\mathbf{X}\mathbf{P}\mathbf{P}^\top\|_F^2 + \|\mathbf{P}\mathbf{P}^\top\mathbf{X}\mathbf{P}\mathbf{P}^\top\|_F^2 + 2\mathrm{tr}\left[(\mathbf{X} - \mathbf{P}\mathbf{P}^\top\mathbf{X}\mathbf{P}\mathbf{P}^\top)\mathbf{P}\mathbf{P}^\top\mathbf{X}\mathbf{P}\mathbf{P}^\top\right].
\end{aligned}$$

It suffices to prove that the trace term is zero:

$$\begin{aligned}
\mathrm{tr}\left[(\mathbf{X} - \mathbf{P}\mathbf{P}^\top\mathbf{X}\mathbf{P}\mathbf{P}^\top)\mathbf{P}\mathbf{P}^\top\mathbf{X}\mathbf{P}\mathbf{P}^\top\right] &= \mathrm{tr}\left(\mathbf{X}\mathbf{P}\mathbf{P}^\top\mathbf{X}\mathbf{P}\mathbf{P}^\top\right) - \mathrm{tr}\left(\mathbf{P}\mathbf{P}^\top\mathbf{X}\mathbf{P}\mathbf{P}^\top\mathbf{P}\mathbf{P}^\top\mathbf{X}\mathbf{P}\mathbf{P}^\top\right) \\
&\overset{(*)}{=} \mathrm{tr}\left(\mathbf{P}^\top\mathbf{X}\mathbf{P}\mathbf{P}^\top\mathbf{X}\mathbf{P}\right) - \mathrm{tr}\left(\mathbf{P}^\top\mathbf{X}\mathbf{P}\mathbf{P}^\top\mathbf{X}\mathbf{P}\right) \\
&= 0.
\end{aligned}$$

Here $(*)$ is due to $\mathbf{P}^\top\mathbf{P} = \mathbf{I}$.

$\square$

**Lemma C.3** (Poincaré separation theorem). *Fix $n \geq m$. Suppose $\mathbf{X}$ is a symmetric $n \times n$ matrix, $\mathbf{P}$ is an $n \times m$ matrix that satisfies $\mathbf{P}^\top\mathbf{P} = \mathbf{I}$, and $\mathbf{Y} = \mathbf{P}^\top\mathbf{X}\mathbf{P}$. Let $\sigma_1(\mathbf{X}) \geq \cdots \geq \sigma_n(\mathbf{X})$ and $\sigma_1(\mathbf{Y}) \geq \cdots \geq \sigma_m(\mathbf{Y})$ be the eigenvalues of $\mathbf{X}$ and $\mathbf{Y}$ in descending order. Then*

$$\sigma_i(\mathbf{X}) \geq \sigma_i(\mathbf{Y}) \geq \sigma_{n-m+i}(\mathbf{X}), \qquad i = 1, \cdots, m.$$

**Lemma C.4** (Weyl's monotonicity theorem). *Suppose $\mathbf{X}, \mathbf{Y}$ are $n \times n$ symmetric matrices, and let $\sigma_1(\mathbf{X}) \geq \cdots \geq \sigma_n(\mathbf{X})$, $\sigma_1(\mathbf{Y}) \geq \cdots \geq \sigma_n(\mathbf{Y})$ and $\sigma_1(\mathbf{X}+\mathbf{Y}) \geq \cdots \geq \sigma_n(\mathbf{X}+\mathbf{Y})$ denote the eigenvalues of $\mathbf{X}, \mathbf{Y}$ and $\mathbf{X}+\mathbf{Y}$ in descending order. Then*

$$\sigma_{i+j-1}(\mathbf{X}+\mathbf{Y}) \leq \sigma_i(\mathbf{X}) + \sigma_j(\mathbf{Y}), \qquad 1 \leq i, j \leq n, i+j-1 \leq n.$$

*In particular, setting $i = 1$ one obtains the commonly used Weyl's inequality: $|\sigma_j(\mathbf{X}+\mathbf{Y}) - \sigma_j(\mathbf{X})| \leq \|\mathbf{Y}\|_2$.*