[Reviews · NeurIPS 2017]

Reviewer 1



This paper shows that given an estimate that is close to the target high-rank PSD matrix A in spectral norm a simple truncated SVD gives an estimate of A that is also close to A in Frobenius norm as well. This leads to a number of applications: high-rank matrix completion, high-rank matrix denoising and low-dimensional estimation of high-dimensional covariance. The paper also gives a bound that depends on the sigma_k - sigma_{k + 1} gap (Theorem 2.2). Overall, I think this is a pretty strong paper that gives a nice set of results. Specific discussion of some of the strengths and weaknesses is below. Strengths: -- Clean set of results, good presentation and clear comparison with the previous work. -- The algorithm is very simple and is very practical. Weaknesses: -- The constant in multiplicative error depends on k and delta -- Results for high-rank matrix completion are conditional on the matrix satisfying a mu_0-spikiness condition which I didn’t find very natural -- No experimental results are given Technical comments: -- In theorem 2.1 you need a bound on delta in terms of \epsilon and \sigma_{k + 1}. How do you ensure such a bound without knowing sigma_{k + 1}?

Reviewer 2



The paper gives an aftermath analysis on the quality of matrix approximation under the Frobenius norm given a bound on the spectral norm. I have capacity to plowing through dense papers so long as they are well put, the results are interesting, and the techniques are novel. I am afraid to say that after 4 pages I felt a strong urge to cease reading the paper. I slogged through the end... The paper does improve the approximation result AM'07. However, the writing is so dense and unfriendly that I wasn't able to distill the gist of it. I will be happy to stand corrected but the paper seems a much better match for ALT / COLT.

Reviewer 3



This paper shows that given an estimate that is close to a general high-rank PSD matrix in spectral norm, the simple SVD of A produces a multiplicative approximation of A in Frobenius norm (Theorem 2.1). This result is discussed with three problems: High-rank matrix completion, High-rank matrix denoising and Low-dimensional approximation of high-dimensional covariance. This paper is well presented, but I would expect more discussions on the results and applications (not only refer to another work). More discussions on related work would be even better.